# Glutathione Supplementation Prevents Neonatal Parenteral Nutrition-Induced Short- and Long-Term Epigenetic and Transcriptional Disruptions of Hepatic H_2_O_2_ Metabolism in Guinea Pigs

**DOI:** 10.3390/nu16060849

**Published:** 2024-03-15

**Authors:** Angela Mungala Lengo, Ibrahim Mohamed, Jean-Claude Lavoie

**Affiliations:** 1CHU Sainte-Justine, Department of Nutrition, Université de Montréal, Montreal, QC H3T 1C5, Canada; angela.mungala.lengo@umontreal.ca; 2CHU Sainte-Justine, Department of Pediatrics-Neonatology, Université de Montréal, Montreal, QC H3T 1C5, Canada; ibrahim.mohamed@umontreal.ca; 3Department of Nutrition, Université de Montréal, Montreal, QC H3T 1J4, Canada

**Keywords:** developmental programming, intravenous GSSG, oxidative stress, neonatal nutrition, nutrigenomics, *GCLC*, *GSase*, *GPx1*, *Nrf2*, *SOD2*

## Abstract

The parenteral nutrition (PN) received by premature newborns is contaminated with peroxides that induce global DNA hypermethylation via oxidative stress. Exposure to peroxides could be an important factor in the induction of chronic diseases such as those observed in adults who were born preterm. As endogenous H_2_O_2_ is a major regulator of glucose–lipid metabolism, our hypothesis was that early exposure to PN induces permanent epigenetic changes in H_2_O_2_ metabolism. Three-day-old guinea pigs were fed orally (ON), PN or glutathione-enriched PN (PN+GSSG). GSSG promotes endogenous peroxide detoxification. After 4 days, half the animals were sacrificed, and the other half were fed ON until 16 weeks of age. The liver was harvested. DNA methylation and mRNA levels were determined for the *SOD2*, *GPx1*, *GCLC*, *GSase*, *Nrf2* and *Keap1* genes. PN induced *GPx1* hypermethylation and decreased *GPx1*, *GCLC* and *GSase* mRNA. These findings were not observed in PN+GSSG. PN+GSSG induced *Nrf2* hypomethylation and increased *Nrf2* and *SOD2* mRNA. These observations were independent of age. In conclusion, in neonatal guinea pigs, PN induces epigenetic changes, affecting the expression of H_2_O_2_ metabolism genes. These changes persist for at least 15 weeks after PN. This disruption may signify a permanent reduction in the capacity to detoxify peroxides.

## 1. Introduction

A growing body of evidence now confirms the relationship between poor dietary quality early in life and an increased risk of chronic diseases associated with impaired glucose and lipid metabolism later in life [1,2,3]. The same risk is observed in adults born extremely preterm [4,5,6,7]. Due to the immaturity of their gastrointestinal tract, these newborns are fed via the parental route. However, their parenteral nutrition (PN) is intrinsically contaminated by peroxides resulting from nutrient auto-oxidation [8,9]. These peroxides are not completely eliminated by the weak antioxidant defense of premature newborns [10,11,12]. Endogenous peroxide (H_2_O_2_) is a powerful regulator of several metabolic pathways based on the oxidation of cysteinyl residues by their proteins. These include glucose and lipid metabolism [13,14,15]. As early as the first week of life, a disturbance in glucose and lipid metabolism associated with PN was observed in premature newborns [14]. Animal studies have shown that the infusion of PN, as received by premature newborns, or of peroxides at levels measured in PN, during their first week of life, induces changes in glucose and lipid metabolism later in life [15]. The biochemical link between neonatal exposure to PN peroxides and the disruption of this metabolism later in life is not fully understood. Verlinden I et al. reported that a PN duration as short as 3 days was sufficient to induce DNA methylation disruption in a pediatric population [16]. In newborn guinea pigs, an infusion of various intravenous solutions containing different levels of oxidizing molecules, such as peroxides, induced a disturbance in the glutathione levels characterized by an oxidation of the redox potential. This oxidation was strongly correlated with the activation of DNA methyltransferase (DNMT) 3a1-2 and overall DNA hypermethylation [17]. Therefore, it was hypothesized that early peroxide exposure could induce a permanent change in the regulation of glucose and lipid metabolism through an epigenetic modification of specific genes. It is well known that epigenetic processes are regulated by endogenous H_2_O_2_ concentration [18,19,20]. We hypothesize that early exposure to PN peroxides induces a permanent epigenetic change in the regulation of genes involved in H_2_O_2_ metabolism. The genes targeted for this study were *SOD2* (superoxide dismutase 2), the enzyme which converts the superoxide anion formed in mitochondria into H_2_O_2_; *GPx1*, which detoxifies H_2_O_2_ to H_2_O; *GCLC* (cysteine ligase) and *GSase* (glutathione synthetase), involved in the synthesis of glutathione; *Nrf2* (nuclear factor erythroid 2-related factor 2), of which the nuclear factor (Nrf2) controls the expression of these four genes; and *Keap1* (Kelch-like ECH-associated protein 1), a regulator of Nrf2 activation.

Thus, the objectives of this study were to investigate the impact of neonatal PN on the methylation of genes involved in endogenous H_2_O_2_ metabolism (Figure 1) and to assess whether these disturbances persist over time. Another objective was to appraise the possible preventive role of glutathione supplementation of PN. Indeed, the main detoxification process of peroxides is the action of glutathione peroxidase (GPx) that uses glutathione (GSH) as a cofactor. However, the glutathione deficiency in premature infants [21,22] limits GPx activity. The low level of glutathione is explained by the limited availability of cysteine, an amino acid constituting glutathione. Indeed, in premature newborns, the conversion of methionine into cysteine is limited by the immaturity of cystathionase, the last enzyme of the transsulfuration pathway [23]. As in premature infants [21,22], PN infusion induces glutathione deficiency in animals [8]. This deficiency was prevented by the addition of disulfide glutathione (GSSG) to their PN [24]. Both GSSG and GSH are substrates for the gamma-glutamyl cycle, which promotes the intracellular transfer of cysteine from plasma glutathione (GSH or GSSG) with a view to the new cellular synthesis of glutathione [25]. GSSG is chosen because of its better stability in PN solutions than GSH [26]. In vivo, glutathione supplementation helps maintain optimal GPx activity to detoxify PN peroxides as they arrive.

The main finding of this study is that neonatal PN induces the hypermethylation of *GPx1* and a lower expression of *GPx1*, *GCLC* and *GSase* at the time of PN administration. The effects persist for at least 15 weeks after stopping PN. Supplementing PN with GSSG prevents all these changes.

## 2. Materials and Methods

### 2.1. Experimental Procedures

The ideal experimental design would include three groups: control animals eating by mouth, animals fed exclusively by infusion of a standard PN known to be contaminated with peroxides, and a third group of animals receiving a peroxide-free PN. However, this PN does not exist and cannot be produced, as the generation of peroxides is caused by the unavoidable interactions between nutrients [8,9,27]. Therefore, the third study group consisted of GSSG-supplemented PN [15,24].

Experimental procedures have been described in previous studies [15,17]. Forty-two male Hartley guinea pigs were purchased from Charles River Laboratories (St-Constant, QC, Canada) on their third day of life. They were housed in the institutional animal facility at constant temperature and humidity as well as 12 h light–12 h dark cycle. Upon arrival, one third received standard guinea pig oral nutrition (ON, control group) and two thirds were fed exclusively by parenteral nutrition (PN group) via a catheter inserted in the jugular vein as previously described [15,17]. Half of animals from PN group received PN solution enriched with disulfide glutathione (PN+GSSG group).

**ON**: animals fed orally ad libitum with regular chow for guinea pigs. (2041-Teklad Global High Fiber Guinea Pig Diet (2.4 kcal/g; 50% energy from carbohydrates; 18% from fat; and 32% from protein, Harlan, Montréal, QC, Canada). These animals ate, on average, 30 g/kg/d of carbohydrates, 10.8 g/kg/d fat and 19.2 g/kg/d proteins.

**PN**: animals fed exclusively with PN containing 8.7% (*w*/*v*) dextrose, 2.0% (*w*/*v*) amino acids (Primene 10%, Baxter, Mississauga, ON, Canada), 1% (*v*/*v*) multivitamin preparation (Multi-12 pediatrics, Sandoz, Montreal, QC, Canada), electrolytes, 1 U/mL heparin and 3.1% (*w*/*v*) lipid emulsion (Intralipid 20%; Pharmacia Upjohn, Baie d’Urfé, QC, Canada). The continuous infusion rate of PN (0.75 kcal/mL) prepared daily was 200 mL/kg/day. These animals received, on average, 7.4 g/kg/d dextrose, 3.2 g/kg/d fatty acids and 4 g/kg/d amino acids.

**PN+GSSG**: animals fed PN supplemented with 12 μM of GSSG (corresponding to 1.3 mg/kg/d) [15,24].

After four days of treatment, at the age of one week, half of the animals in each group were sacrificed for liver harvesting. In the remaining half, the catheters of the PN and PN+GSSG groups were knotted, and animals received standard guinea pig chow until the 16th week of life. At 16 weeks, the liver was harvested and stored at −80 °C until gene methylation levels could be determined. To document the biological impact of these methylation changes, the gene mRNA levels were also measured.

The study was approved by the Institutional Committee for Good Practice with Animals in Research of the CHU Sainte-Justine in accordance with the Canadian Council on Animal Care guidelines (protocol #672).

### 2.2. Biochemical Assessments

#### 2.2.1. DNA and RNA Extraction

Total RNA and genomic DNA was extracted using about 30 mg of liver and purified using the AllPrep DNA/RNA/miRNA kit (# 80224, Qiagen, Toronto, ON, Canada) according to the manufacturer’s protocol. RNA and DNA concentrations and purity were measured by BioTek Epoch Microplate Spectrophotometer (Agilent Technologies, Mississauga, ON, Canada); optical density at 260/280 nm ranged from 1.8 to 2.0. RNA integrity was analyzed using RNA ScreenTape analysis (Agilent Technologies, Mississauga, ON, Canada).

#### 2.2.2. Methylation Analysis

Gene-specific DNA methylation (5-Methylcytosine) was quantified using the Methylated DNA Immunoprecipitation (MeDIP)—quantitative PCR (qPCR) analysis, as described by Karpova et al. [28]. CpG islands located in the promoter, TSS and first exon regions were analyzed.

##### Promoter Analysis and Primer Design

The Ensembleast genome browser (https://useast.ensembl.org/) (accessed from 12 April 2019 to 24 November 2022) was used to retrieve all the targeted gene’s sequences, except for the Gpx1 gene, for which we used the NCBI genome database (Gpx1gene ID: 100729115). Using the Methprimer tool (https://www.urogene.org/cgi-bin/methprimer/methprimer.cgi), a CpG-rich region closed to first exon and transcription start sites was identified (−2000 to +1000 pb) then screened for promoter elements using the genome regulation analysis online tool (http://www.softberry.com) or manually to validate the presence of at least three promoters’ elements in the core sequence. Based on these screenings, primers were designed for one or two amplicons (Table 1) with the Primer3web online software, Version 4.1.0. Primers were ordered from Integrated DNA Technologies (IDT, Coralville, IA, USA). All primer sequences were validated beforehand using the IDT OligoAnalyzer™ Tool to ensure that only a single product was obtained. Based on sequence screening and specific features of promoter regions of *GCLC* and *SOD2*, methylation analyses were performed using two amplicons for each gene. For *GCLC*, amplicon #1 covers the region which extends from −366 bp to −212 bp from TSS and amplicon #2 from +25 bp to +139 bp. Regarding *SOD2*, amplicon #1 was designed between −158 bp to −5 bp from TSS, and amplicon #2 from −29 bp to +203 bp.

##### Immunoprecipitation of Genomic DNA

Briefly, 18–20 μg of purified DNA were diluted in 500 μL TE buffer (Tris-EDTA: 10 mM Tris–HCl, 1 mM EDTA pH 8.0, stored at 4 °C) in a 2 mL tube. Then, 50 μL of diluted DNA was kept aside for a non-sonicated control. The rest of the diluted sample was sonicated on ice for 20 cycles, 30 s ON/30 s OFF, with 60% amplitude using a Vibra-Cell Ultrasonic Liquid Processor (Sonics & Materials Inc., Newtown, CT, USA). To check the efficiency of sonication, 10 μL of the non-sonicated control and 10 μL of the sonicated sample were loaded on 1.3% agarose gel and run at 100 V for 70 min. The size of the DNA fragments ranged from 200 to 1000 bp. Sonicated DNA samples were denatured at +100 °C for 10 min and immediately cooled on ice for at least 5 min. For immunoprecipitation (IP) of methylated DNA, 1 μg (30 μL) of sonicated and denatured DNA sample was mixed with 20 μL of 10 x IP buffer (100 mM sodium phosphate buffer pH 7.0, 1.4 M NaCl, 0.5% Triton X-100), 2 μL of anti-5mC antibody (Cell Biolabs, San Diego, CA, USA) and RNase free water up to 200 μL. For the negative control IP, two tubes were prepared: same mix (i) without antibody/with DNA samples, and (ii) with antibody/without DNA sample. All tubes were incubated overnight in a cold room (4 °C) with overhead shaking. Each IP sample and negative control tube was mixed by inversion with 20 μL of pre-washed Dynabeads™ Protein G (Invitrogen by ThermoFisher Scientific, Waltham, MA, USA), then incubated for 2 h at +4 °C with overhead agitation. Immuno-precipitated methylated DNA-beads complex were collected on a magnetic rack and washed three times with 400 μL of IP buffer, then resuspended in a freshly prepared protein digestion solution (200 μL of Proteinase K digestion buffer and 2.5 μL Proteinase K (20 μg/mL, # 19131, Qiagen, Germantown, MD, USA). Tubes were incubated for 2 h at 56 °C, vortexed every 15 min, then incubated a second time for 30 min at 95 °C. The IP methylated DNA was released from the beads using a magnetic holder, purified using the column-based PCR cleanup kit (Bio-Basic, Markham, ON, Canada), according to the manufacturer’s protocol, and DNA was eluted with 40 μL 10 mM Tris pH 8.0.

##### qPCR Analysis of Immunoprecipitated DNA

Real-time PCR was performed by amplifying 2.5 μL of IP methylated DNA samples or input samples (non-sonicated, denatured and non-IPed DNA), by using 400 to 700 μM of forward and reverse primers for each targeted gene, and an appropriate volume of iTaq Universal SYBR Green Supermix (Bio-Rad Laboratories, Mississauga, ON, Canada) in the LightCycler^®^ 96 (Roche Diagnostics, IN, USA). The thermal cycling conditions included initial denaturation at 95 °C for 200 s and subsequent 40 cycles of a 2-step amplification protocol: 95 °C for 15 s and 60/62 °C for 60 s. The 2-step protocol was followed by melting: 95 °C for 10 s, 65 °C for 60 s and 97 °C for 1 s. The relative methylation levels were calculated from their quantification cycle values (Cq) by normalizing with the Cq values of input samples, using the comparative threshold method (2-ΔCT) [29].

##### Gene Expression Analysis by RT-qPCR

Purified RNA was treated with DNAse I, RNase-free (Thermo Scientific™, Waltham, MA, USA) according to the manufacturer’s protocol, and reverse transcribed into cDNA using the iScript™ Reverse Transcription Supermix (Bio-Rad Laboratories, Mississauga, ON, Canada). Briefly, 1 μg of DNAse I-treated RNA was mixed with 4 μL of iScript RT Supermix and 4 μL of nuclease-free water. The reaction mix was incubated in a thermal cycler using the manufacturer’s recommended protocol. The fresh cDNA was amplified with 150–500 μM of forward and reverse primers for each targeted gene (Table 2) and an appropriate volume of iTaq Universal SYBR Green Supermix (Bio-Rad Laboratories, Mississauga, ON, Canada) in the LightCycler^®^ 96 (Roche Diagnostics, IN, USA) with the thermal cycling conditions described previously. 

The relative mRNA expression was calculated from the quantification cycle values (Cq) of targeted genes by normalizing with the Cq values of the reference gene, glyceraldehyde 3-phosphate dehydrogenase (GAPDH), using the comparative threshold method (2^−ΔCT^) [29].

### 2.3. Statistics

Data were orthogonally compared by factorial ANOVA after verification of homoscedasticity by Bartlett’s χ^2^. *SOD2* mRNA data were transformed by the natural logarithm to achieve homoscedasticity. The ON vs. PN comparison aims to answer the question of whether early an infusion of PN affects the methylation of at least one gene. To test whether this change is induced by peroxide-associated oxidative stress, the PN+GSSG group was used.

An absence of statistical interaction between age and treatments would suggest that the effects observed during the treatments persist at least until 15 weeks after the cessation of treatment. All results are expressed as mean ± Standard Error of the Mean (SEM) and as raw data. The threshold of significance was set at *p* < 0.05.

## 3. Results

### 3.1. SOD2 and GPx1

Figure 2 groups together the results for *SOD2* (the enzyme that generates H_2_O_2_) and *GPx1* (the enzyme that detoxifies peroxides).

To assess the methylation levels for *SOD2*, two regions containing CpG islands located within the promoter regions were amplified. DNA methylation levels (Figure 2A,B) did not vary between groups, either for each amplicon or for all amplicons (*p* > 0.23), nor between ages (*p* = 0.09). There was no interaction between age and treatments (*p* > 0.63).

PN did not significantly alter the mRNA expression of the *SOD2* (Figure 2C) compared to the ON group (*p* = 0.25). However, GSSG supplementation up-regulated (*p* < 0.01) the mRNA compared to the ON and PN groups. In the absence of significant interaction (*p* = 0.06), these observations also occurred at 16 weeks. The average impact of GSSG supplementation in PN was an increase of 54% at 1 week and 21% at 16 weeks of life. The expression of mRNA was higher in adult animals (*p* < 0.001).

As shown in Figure 2D, animals given PN during their first week of life had significantly higher methylation levels of *GPx1* compared to the ON and PN+GSSG groups (*p* < 0.05), while no significant difference was observed between the ON and PN+GSSG groups (*p* = 0.72). There was no difference between ages (*p* = 0.19).

Although there was no significant difference in *GPx1* mRNA levels (Figure 2E) between the ON and PN+GSSG groups (*p* = 0.83), these were lower in the PN groups (*p* < 0.05). There was no difference between ages (*p* = 0.08).

The absence of interaction between treatments and ages (*p* > 0.5) suggested that the DNA methylation and mRNA profiles observed at 1 week of age persisted over time, at least until 15 weeks after the cessation of treatments. However, the average impact of PN peroxides at one week was a 76% increase in methylation and a 25% decrease in mRNA, while at 16 weeks it was a 113% increase in methylation and a 10% decrease in mRNA.

### 3.2. GCLC and GSase

Figure 3 groups together the results for *GCLC* and *GSase.* The genes encoding the two enzymes involved in the synthesis of glutathione.

The *GCLC* methylation levels in the TATA box region (amplicon #1) (Figure 3A) presented a significant (*p* = 0.02) interaction between age and treatments. In this case, the ANOVA was repeated for each age. There was no statistical difference between treatments for each age (*p* > 0.18). The analysis of CpGs-rich amplicon #2 (Figure 3B), covering the TSS and first exon region of *GCLC*, showed no difference between treatments (*p* = 0.62) or ages (*p* = 0.12), and no interaction (*p* = 0.53).

For *GCLC* mRNA (Figure 3C), there was no significant interaction between treatments and ages (*p* = 0.06) and no difference between ages (*p* = 0.30). Nonetheless, PN induced a lower mRNA level (*p* < 0.01) compared to the ON and PN+GSSG groups, which had similar values (*p* = 0.84). The average impact of PN peroxides was a 40% decrease at 1 week of age, while at 16 weeks of age, a 9% decrease was observed.

The methylation analysis of the amplicon used for *GSase* (Figure 3D) showed no interactions between treatments and ages (*p* = 0.87) or between treatments (*p* = 0.24). However, the methylation levels could be lower at 16 weeks as the *p*-value is 0.0503.

Regarding *GSase* mRNA levels (Figure 3E), there was no significant interaction between treatments and ages (*p* = 0.19), and no difference between ages (*p* = 0.64). PN induced a lower mRNA expression (*p* < 0.001) compared to the ON and PN+GSSG groups, which had similar values (*p* = 0.10). The average impact of PN peroxides was a 40% decrease at 1 week, while at 16 weeks it was a 19% decrease.

### 3.3. Nrf2 and Keap1

Figure 4 groups the results for *Nrf1* and *Keap1*. The latter is the gene encoding the Nrf2 regulatory protein.

The quantification of the DNA methylation of *Nrf2* (Figure 4A) in a CpGs-rich amplicon revealed no difference between the ON and PN groups (*p* = 0.57), while GSSG supplementation in PN attenuates DNA methylation levels when compared to ON and PN groups (*p* = 0.03). There was no difference between ages (*p* = 0.11) and no interaction between groups (*p* = 0.62).

There was no significant difference in *Nrf2* mRNA (Figure 4B) between the PN compared to the ON group (*p* = 0.86), while we observed a significant up-regulation in the PN+GSSG groups (*p* < 0.001), compared to the ON and PN groups. Also, there was no significant interaction between treatments and ages (*p* = 0.34). Animals that were 16 weeks old had 50% higher mRNA levels (*p* = 0.001). The average impact of GSSG supplementation in PN at 1 week was a 48% decrease in methylation and a 112% increase in mRNA, while at 16 weeks, a 43% decrease in methylation and a 31% increase in mRNA were measured.

In the case of *Keap1*, levels of methylation (Figure 4C) did not vary between the treatments (*p* = 0.11) and no interaction was observed (*p* = 0.65). However, a significant diminution of methylation levels at 16 weeks was observed (*p* = 0.01).

*Keap1* mRNA levels (Figure 4D) were lower in the PN and PN+GSSG groups compared to the ON group (*p* = 0.03). Values were similar between the PN and PN+GSSG groups (*p* = 0.46). There was no difference according to ages (*p* = 0.48) and no significant interaction (*p* = 0.24). However, the average impact of PN and PN+ GSSG was a 27% decrease at 1 week, while at 16 weeks it was a 5% decrease.

## 4. Discussion

The study confirms that, at least in guinea pigs, PN received early in life can alter endogenous H_2_O_2_ metabolism through epigenetic modification, and that this change is maintained over time, at least 15 weeks after PN is discontinued. The prevention achieved by glutathione supplementation of PN suggests that the epigenetic modification was caused, at least in part, by peroxides contaminating PN. Here, epigenetic modification refers to DNA methylation, which can lead to the suppression of gene expression [30].

A change in methylation levels was only observed for two genes, while the mRNA levels of all six studied genes were affected. Compared with the ON (control) groups, PN induced a higher level of *Gpx1* methylation, while GSSG added to PN induced a lower level of Nrf2 methylation. Specifically, for *Gpx1*, the addition of GSSG prevented PN-induced gene methylation. These results are consistent with the concept that the addition of GSSG to PN improves tissue GSH levels, which helps detoxify peroxides [24]. Nguyen et al. [31] also reported the hypermethylation and downregulation of *GPx1* in ATX mice, a model used to study the oxidative environments associated with dyslipidemia and atherosclerosis. In contrast, *Nrf2* methylation was resistant to PN, suggesting that the methylation process of this gene is independent of PN-infused peroxides. Surprisingly, the supplementation of PN with GSSG induced the demethylation of this gene. One might think that this supplementation promoted a higher-than-normal level of GSH, which could have induced an abnormal reduction in endogenous peroxides. However, previous studies conducted with the same animal model and using the same or a similar concentration of GSSG in PN demonstrated that this supplementation restored, not increased, tissue glutathione levels compared to those measured in control animals [15,24]. This effect must be explained by another mechanism. One possible explanation is that the correction of glutathione deficiency with GSSG supplementation enabled the activity of other PN nutrients, such as ascorbate, an important player in DNA demethylation [32]. Future studies could elucidate this phenomenon. Nevertheless, this demethylation activity induced by GSSG supplementation may also explain the preventive effect on *Gpx1* methylation observed in the PN+GSSG groups. PN induces hypermethylation and GSSG supplementation induces demethylation, resulting in a methylation level similar to that of the control group. Whatever the mechanism, the fact that GSSG supplementation prevents *Gpx1* hypermethylation is important. Indeed, GSSG supplementation is proposed for premature newborns who require PN [8,24,26].

One biological consequence of DNA methylation is altered gene expression. PN has been associated with higher *GPx1* methylation and lower mRNA levels. On the other hand, the mRNA levels of *GCLC* and *GSase*, the two genes encoding enzymes involved in glutathione synthesis, were lower in the PN groups than in the ON groups, and this without observing any change in their methylation levels. This may be due to an error in the choice of DNA sequences, or to the involvement of another epigenetic mechanism such as histone methylation. The similarity between the results obtained at 1 and 16 weeks of age suggests that the mRNA modifications may be related to an epigenetic phenomenon. These lower mRNA levels could translate into lower protein levels. These findings, therefore, suggest a reduced ability to detoxify peroxides, including endogenous H_2_O_2_, in people who receive or have received PN early in life. A deficiency in or the reduced activity of GPx1 have been associated with an increased risk of cardiovascular diseases [33,34], diabetes [35] and metabolic syndrome [36]. The good news is that the enrichment of PN with GSSG prevented these changes in both the *GPx1* methylation and mRNA levels of these three genes. *SOD2* and *Nrf2* gene methylation and mRNA levels were not affected by PN but were by GSSG supplementation. In the PN+GSSG groups, their methylation levels were lower and mRNA levels higher. Higher *SOD2* activity could lead to lower superoxide anion concentration and higher H_2_O_2_ production. With efficient *GPx1* activity, this increase in H_2_O_2_ would be limited to maintain homeostasis of its concentration, while lowering the level of the free radical superoxide anion. Similarly, a higher level of Nrf2 could translate into greater efficiency in stimulating the transcription of genes encoding antioxidant enzymes. Both elements represent an advantage in the control of oxidative stress throughout life.

*Keap1* is a key factor in the regulation of Nrf2 activation. PN, independently of the presence of GSSG, induced a lower level of Keap1 mRNA, with no change in methylation levels. The lack of impact on methylation can be explained by the fact that 95% of the *Keap1* promoter region in guinea pigs is not established. With this specific conformation, we may have missed the methylation zone that regulates gene expression. The lower level of *Keap1* mRNA in the PN groups is similar to that observed for *GPx1*, *GCLC* and *GSase* mRNAs. The difference is the absence of a GSSG effect. Based on the previous suggestion that GSSG in PN could promote gene demethylation (see for Nrf2), the absence of a GSSG effect might suggest that the modulation of *Keap1* mRNA level by PN was not conditioned by gene methylation. A lower level of *Keap1* mRNA, even slightly, over the course of life, could translate into a greater facility to induce Nrf2 activation.

The lack of statistical interaction between treatments and ages suggests that, for all genes studied, methylation and expression changes persist over time, at least until 15 weeks after PN cessation. However, the amplitude of mRNA changes was lower at 16 weeks. For example, the average impact of PN peroxides on *GPx1*, *GCLC* and *GSase* mRNA was a 35% and 13% decrease at 1 and 16 weeks, respectively. Similarly, the effect of GSSG supplementation on *SOD2* and *Nrf2* mRNA was an 83% increase at 1 week and a 26% increase at 16 weeks. These observations contrast with the methylation changes that were similar between ages (for *GPx1* and *Nrf2*). Thus, the induced methylation changes could be permanent, while, over time, general metabolism could tend to compensate for the impact of epigenetic disorders induced by early exposure to PN peroxides. It would be desirable for a future study to compare these epigenetic and gene expression variations at different times of life. Has the extent of mRNA modification reached a plateau? If so, the magnitude of these changes could be sufficient to induce metabolic alterations, the clinical manifestations of which will only appear later in life.

## 5. Conclusions

Although parenteral nutrition is essential for premature newborns, it is associated with several health complications. The risk of developing chronic metabolic diseases during their lifetime is high. The current study suggests that the epigenetic disruption of endogenous H_2_O_2_ metabolism may be one of the causes. This phenomenon was demonstrated here using a term animal model. We believe that the window of sensitivity is independent of gestational age at birth. Prematurity can only be an aggravating factor in these changes, since glutathione levels are very low in these infants [21,22]. Verlinden I et al. reported that PN induced DNA methylation disruption in the 2- to 4-year-old pediatric population, independent of gestational age at birth [16]. A recent workshop (2018) by the American Society for Parenteral and Enteral Nutrition (ASPEN) reported that the quality of nutrition in early childhood and exposure to xenobiotics could increase the risk of developing metabolic diseases such as diabetes, obesity, hypertension, cardiovascular diseases, and even mental illness [37]. PN is a source of xenobiotics, such as H_2_O_2_, lipid aldehydes and organic peroxides [8,9,38,39], all of which interfere with glutathione, thus weakening the child’s antioxidant defense. Several studies indicate that oxidative stress or altered H_2_O_2_ metabolism [40,41] are inducers of chronic diseases in adults, such as diabetes, cardiovascular disease and metabolic disorders [42]. The good news is that, as shown here with neonatal guinea pigs, the glutathione supplementation of PN can prevent the PN-induced short- and long-term disruption of H_2_O_2_ metabolism.

## Figures and Tables

**Figure 1 nutrients-16-00849-f001:**
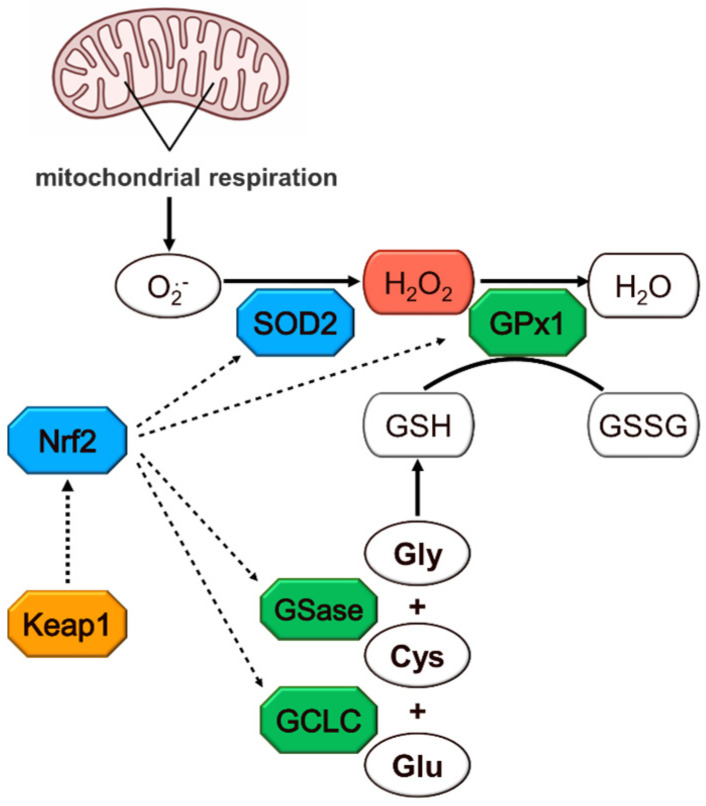
Hydrogen peroxide metabolism. Colors indicate the enzymes whose genes were investigated for their methylation and expression levels. Cys: cysteine; GCLC: glutamate–cysteine ligase catalytic subunit; Gly: glycine; Glu: glutamate; GSH: glutathione; GSase: glutathione synthase; GSSG: glutathione disulfide; GPx1: glutathione peroxidase 1; H_2_O_2_: hydrogen peroxide; Keap1: Kelch-like ECH-associated protein 1; Nrf2: nuclear factor erythroid 2-related factor 2; O^•^_2_^−^: superoxide anion radical; SOD2: superoxide dismutase 2.

**Figure 2 nutrients-16-00849-f002:**
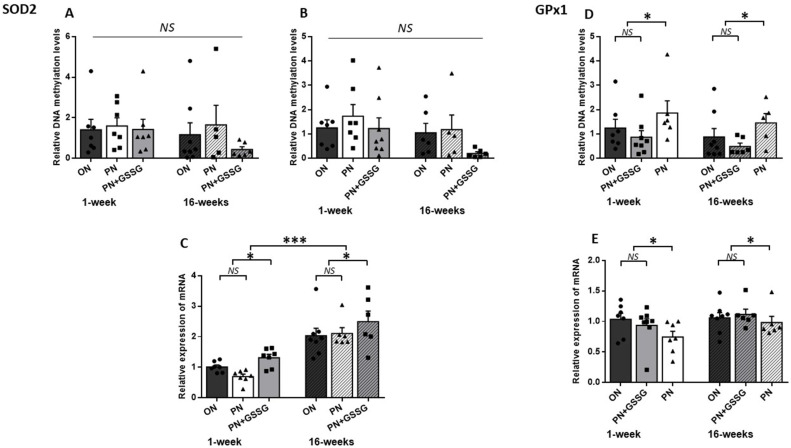
DNA methylation and mRNA levels for *SOD2* and *GPx1*. Panels (**A**,**B**): DNA methylation levels in the *SOD2* promoter region with Amplicon #1 and Amplicon #2, respectively. No statistical difference between groups. Panel (**C**): *SOD2* mRNA levels. There was no statistical difference between ON and PN, both being lower than the PN+GSSG groups. Overall, levels were higher in animals 15 weeks after cessation of treatment. Panel (**D**): DNA methylation levels in the promoter region of *GPx1*. No statistical difference between ON and PN+GSSG groups; both were lower than PN groups. There was no difference between ages. Panel (**E**): *GPx1* mRNA levels. No statistical difference between ON and PN+GSSG groups; both were higher than PN groups. There was no difference between ages. ON: Orally fed animals. PN: animals receiving parenteral nutrition. PN+GSSG: animals with PN enriched with GSSG. Means ± SEM, n = 5–8. NS: no statistical significance; *: *p* < 0.05; ***: *p* < 0.001.

**Figure 3 nutrients-16-00849-f003:**
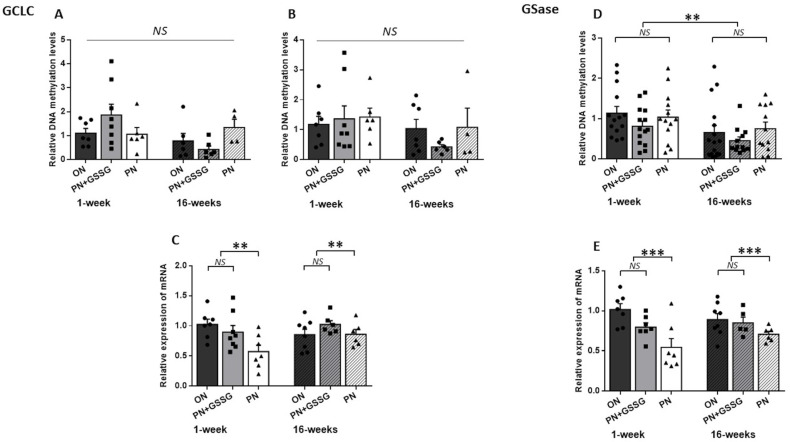
DNA methylation and mRNA levels for *GCLC* and *GSase.* Panels (**A**,**B**): DNA methylation levels in the *GCLC* promoter region with Amplicon #1 and Amplicon #2, respectively. There was no statistical difference between treatments. Panel (**C**): *GCLC* mRNA levels. There was no statistical difference between ON and PN+GSSG, both being higher than the PN groups. Panel (**D**): DNA methylation levels in the promoter region of *GSase*. No statistical difference between treatments. The levels were lower at 16 weeks. Panel (**E**): *GSase* mRNA levels. No statistical difference between ON and PN+GSSG groups; both were higher than PN groups. There was no difference between ages. ON: orally fed animals. PN: animals receiving parenteral nutrition. PN+GSSG: animals with PN enriched with GSSG. Means ± SEM, n = 5–8. NS: no statistical significance; **: *p* < 0.01; ***: *p* < 0.001.

**Figure 4 nutrients-16-00849-f004:**
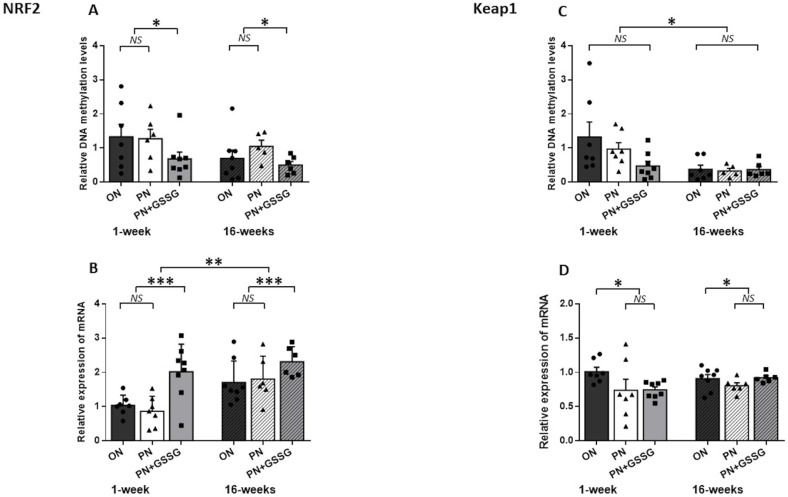
DNA methylation and mRNA levels for *Nrf2* and *Keap1.* Panel (**A**): DNA methylation levels in the promoter region of *Nrf2*. No difference between ON and PN groups; both were higher than PN+GSSG groups. There was no difference between ages. Panel (**B**): *Nrf2* mRNA levels. No difference between ON and PN groups; both were lower than PN+GSSG groups. Levels were higher at 16 weeks. Panel (**C**): DNA methylation levels in the promoter region of *Keap1*. No difference between treatments. Levels were lower at 16 weeks. Panel (**D**): *Keap1* mRNA levels. No difference between PN and PN+GSSG groups; both were lower than ON groups. There was no interaction or difference between ages. ON: orally fed animals. PN: animals receiving parenteral nutrition. PN+GSSG: animals with PN enriched with GSSG. Means ± SEM, n = 5–8. NS: no statistical significance; *: *p* < 0.05; **: *p* < 0.01; ***: *p* < 0.001.

**Table 1 nutrients-16-00849-t001:** Primer sequences and concentrations used in IP RT-qPCR experiments.

Genes	Amplicon from TSS	IP-PCR Primers	Product (pb)	Concentration Used (nM)
*Nrf2*	+417 to +637	5’-CGGACCACACTGAGGACTTG-3’	221	500
		5’-CGCAACAGATCAACAGCTCC-3’		
*Keap1*	+1228 to +1478	5’-CTACAACCCCATGACCAACCAG-3’	250	400
		5’-TGTCTCAAAACACCAAACTCAGC-3’		
*SOD2*	−158 to −5	5’-CAGTGGGATAAAGTGAGCCTGG-3’	153	450
Amplicon 1		5’-AGCGTGTTAGTGGAGGGTTTC-3’		
*SOD2*	−29 to +203	5’-CCCGAAACCCTCCACTAACAC-3’	233	700
Amplicon 2		5’-CAGTTGGTATGGTCCGCAGC-3’		
*GCLC*	−366 to −212	5’-CCTCCGCTCCTTGGACCGTC-3’	154	700
Amplicon 1		5’-CTACTTTGTGACGAGAAGGCTGC-3’		
*GCLC*	+25 to +139	5’-CGCTGAACTGGGAGGAGACC-3’	114	500
Amplicon 2		5’-CACTTGAGCACGTCCTTCTGC-3’		
*Gase*	+307 to +508	5’-GCCCTACTTCTTAAACCCGCT-3’	201	500
		5’-AGACCCCTCCCTCACATCTT-3’		
*GPx1*	+74 to +236	5’-GCTTTCGTCATGTGTGCTGC-3’	163	600
		5’-GGACCGTGGTGCCTCAAAG-3’		

**Table 2 nutrients-16-00849-t002:** Forward and reverse primers sequences for each targeted gene.

Genes	Primers Sequences	
	Forward	Reverse
*Nrf2*	GCTAGATGAAGAGACAGGGGA	ACAAATGGGAATGTTTCTGCCA
*Keap1*	TGCTACAACCCCATGACCAA	ACCAAGTGCCACTCGTCC
*SOD2*	CTACGACTATGGGGCCCTAC	CACCGTTGAACTTCAGTGCA
*GCLC*	TGGGGAGAAGTACAACGACA	GGCATCATCCAGGTCGATCT
*Gase*	ACTGTGTTCCTGGGCTTGAA	CTTGGCAGCTTCTTCAGTCC
*GPx1*	TTGAGAATGTGGCCTCCCTT	CGGACGTACTTGAGCGAATG
*GAPDH*	GATCCCGCCAACATCAAGTG	CACGCCCATCACGAACATAG

## Data Availability

Data are contained within the article.

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
