# Peer review of "Glutathione Supplementation Prevents Neonatal Parenteral Nutrition-Induced Short- and Long-Term Epigenetic and Transcriptional Disruptions of Hepatic H2O2 Metabolism in Guinea Pigs"

_nutrients, 2024, doi:10.3390/nu16060849_

Round 1
Reviewer 1 Report
Comments and Suggestions for Authors
The authors measured gene methylation and expression for antioxidative enzyme chain in neonatal guinea pigs fed either by mouth or by standard PN and PN supplemented with oxidised glutathione (GSSG).
Autos found changes in DNA methylation and RNA expression after parenteral nutrition period and (surprisingly) 16 weeks after PN and these changes were positively influenced by GSSG. These experimental results are well documented.
Comment: Are authors able to document any biochemical or metabolic consequence of changes in gene expression and DNA methylation? These results will improve informational and cost of the article.
Author Response
Manuscript Nutrients-2819523 entitled: ”Glutathione supplementation prevents neonatal parenteral nutrition induced short- and long-term epigenetic and transcriptional disruptions of hepatic H2O2 metabolism in guinea pigs.”
Authors: Angela Mungala Lengo, Ibrahim Mohamed, Jean-Claude Lavoie *
Answers to the Editor and Reviewers.
Comment (C), Answer (A)
Reviewer 1:
Authors found changes in DNA methylation and RNA expression after parenteral nutrition period and (surprisingly) 16 weeks after PN and these changes were positively influenced by GSSG. These experimental results are well documented.
C: Are authors able to document any biochemical or metabolic consequence of changes in gene expression and DNA methylation? These results will improve informational and cost of the article.
A: Although we agree with these comments, we cannot respond to them. I explained to the editor our impossibility. Ms Kimprin Zhang suggested that I forward my explanation to you. Here it is:
“I understand the reviewer's request. Of course, the addition of Western blot and enzyme activity data would be appropriate. This would consolidate the results reported in the manuscript.
Unfortunately, this is not possible for us. I'm retired. My laboratory is closed. The last member of my team, Ms Angela Mungala Lengo (first author of the manuscript), will submit her Ph.D. thesis earlier this week.
We thought that our results demonstrating that an unavoidable mode of nutrition (parenteral) in very premature newborns induces a permanent epigenetic modification of endogenous H2O2 metabolism were important enough to be published in the journal Nutrients.
The fact that this diet is associated with oxidative stress (oxidation of redox potential) is well known and frequently reported in the scientific literature. The relationship between oxidative stress induced by parenteral nutrition and the enzymatic mechanisms leading to DNA methylation is also well documented (Mungala Lengo A, et al. Relationship between redox potential of glutathione and DNA methylation level in liver of newborn guinea pigs. Epigenetics, 15(12): 1348-1360, 2020). The manuscript goes on to explain the importance of these results and shows that this mode of nutrition could have serious biological consequences, as it could affect permanently methylation of genes involved in H2O2 metabolism, which is an important regulator of glucose and lipid metabolism. Other teams can easily study the biological impact of our findings.
…
Jean-Claude Lavoie“
Thank you for helping us improve the manuscript.
Reviewer 2 Report
Comments and Suggestions for Authors
The authors describe the effect of oxidative stress from peroxides in parenteral nutrition on changes to DNA methylation and gene expression of a select number of genes involved in the glutathione pathway. The study design with animal groups two time points for both short-term and long-term effects are positives for this manuscript. However, this manuscript is lacking in depth and clarity.
General Comments:
- This paper needs to be revised with a native English speaking editor. There are multiple subject/verb agreement issues and in some cases sentences that do not make any sense.
- Given the impact factor of Nutrients, there is insufficient analysis provided to consider this a complete manuscript. The only data is methylation and mRNA for 6 genes. The authors provide no information on oxidative stress status in the animals. There are numerous kits that can be purchased from companies to measure oxidative stress in serum. The author's also don't provide any activity for the genes, there are kits for SOD, GPx, etc... The study also goes into detail about the effect that PN has on glucose metabolism, anything to corroborate this data to connect to the purpose of your current analysis on oxidative stress markers would be appropriate. Larger scale analysis could also have been performed. RNA-seq with pathway analysis for oxidative stress pathways to show global changes between groups would be stronger data, followed by the methylation data on the 6 genes would be stronger results.
Methods:
- The methods are not well-written in general. Under experimental procedures, the authors define the ON diet by percent energy, but do not give this for the PN; rather, they provide w/v. Further, they provide GSSG as uM. They need to use a consistent set of standards to provide clear information on what the animals received.
- The authors provide a significant amount of information on how the genomic DNA was isolated but offer very little in the methods in the section "Methylation analysis". Given this, is a major component of each figure, they should give a more detailed explanation of this protocol.
-The statistics are written in an unusual way. A lot of text is given to "what" the meaning of each difference would represents This really isn't appropriate for the methods section.
Results:
- All of the figures could be represented as a single figure with multiple panels.
- The presentation of the statistics on the graphs are somewhat confusing. I am not clear what the bar without the asterisks are represent. I would remove those and only show bars connecting significantly different groups. Also, the x-axis should have a line on it.
- In the figure legends, typically definitions to abbreviations are at the end of the legend. There is also discussion of results within the legend. I do not think this is necessary.
Comments on the Quality of English LanguageThere are numerous issues with language. This work needs editing by a native English speaker. This problem is in combination with poor error checking for sentences that were edited that no longer make any sense. The second sentence of the abstract is an example of this.
Author Response
Manuscript Nutrients-2819523 entitled: ”Glutathione supplementation prevents neonatal parenteral nutrition induced short- and long-term epigenetic and transcriptional disruptions of hepatic H2O2 metabolism in guinea pigs.”
Authors: Angela Mungala Lengo, Ibrahim Mohamed, Jean-Claude Lavoie *
Answers to the Editor and Reviewers.
Comment (C), Answer (A)
Reviewer 2:
The authors describe the effect of oxidative stress from peroxides in parenteral nutrition on changes to DNA methylation and gene expression of a select number of genes involved in the glutathione pathway. The study design with animal groups two time points for both short-term and long-term effects are positives for this manuscript. However, this manuscript is lacking in depth and clarity.
C1: The author's also don't provide any activity for the genes, there are kits for SOD, GPx, etc… The study also goes into detail about the effect that PN has on glucose metabolism, anything to corroborate this data to connect to the purpose of your current analysis on oxidative stress markers would be appropriate. Larger scale analysis could also have been performed. RNA-seq with pathway analysis for oxidative stress pathways to show global changes between groups would be stronger data, followed by the methylation data on the 6 genes would be stronger results.
A: Although we agree with these comments, we cannot respond to them. I explained to the editor our impossibility. Ms Kimprin Zhang suggested that I forward my explanation to you. Here it is:
“I understand the reviewer's request. Of course, the addition of Western blot and enzyme activity data would be appropriate. This would consolidate the results reported in the manuscript.
Unfortunately, this is not possible for us. I'm retired. My laboratory is closed. The last member of my team, Ms Angela Mungala Lengo (first author of the manuscript), will submit her Ph.D. thesis earlier this week.
We thought that our results demonstrating that an unavoidable mode of nutrition (parenteral) in very premature newborns induces a permanent epigenetic modification of endogenous H2O2 metabolism were important enough to be published in the journal Nutrients.
The fact that this diet is associated with oxidative stress (oxidation of redox potential) is well known and frequently reported in the scientific literature. The relationship between oxidative stress induced by parenteral nutrition and the enzymatic mechanisms leading to DNA methylation is also well documented (Mungala Lengo A, et al. Relationship between redox potential of glutathione and DNA methylation level in liver of newborn guinea pigs. Epigenetics, 15(12): 1348-1360, 2020). The manuscript goes on to explain the importance of these results and shows that this mode of nutrition could have serious biological consequences, as it could affect permanently methylation of genes involved in H2O2 metabolism, which is an important regulator of glucose and lipid metabolism. Other teams can easily study the biological impact of our findings.
…
Jean-Claude Lavoie“
C2: The authors provide no information on oxidative stress status in the animals. There are numerous kits that can be purchased from companies to measure oxidative stress in serum.
A: The relationship between parenteral nutrition (PN) and oxidative stress, characterized by the oxidation of glutathione redox potential, is well documented. This is reported by several literature reviews, including the one cited in reference 9. However, the manuscript often refers to oxidative stress without documenting it. The main objective of the manuscript being to study the impact of PN on the methylation of hydrogen peroxide metabolism genes, taking the opportunity to correct the quality of English, any reference to oxidative stress induced by PN has been removed from the manuscript. The important message is that PN administered to premature infants induces epigenetic modifications that appear to be permanent and that GSSG enrichment prevents these modifications. The purpose of the manuscript was not to provide all the metabolic explanations that could explain these results. Now, only, because of the use of GSSG, the possible involvement of peroxides from PN is mentioned.
C3: The methods are not well-written in general. Under experimental procedures, the authors define the ON diet by percent energy, but do not give this for the PN; rather, they provide w/v. Further, they provide GSSG as uM. They need to use a consistent set of standards to provide clear information on what the animals received.
A: Consistent with this comment, additional information has been added in Section 2.1. Experimental procedures. At the end of the paragraph beginning with “ON” the following statement was added: “These animals ate on average 30 g/kg/d of carbohydrates, 10.8 g/kg/d fat and 19.2 g/kg/d proteins.”
In the fifth line of the paragraph beginning with “PN”, the sentence “The continuous infusion of PN prepared daily …” was modified to “The continuous infusion of PN prepared daily (0.75 kcal/mL)…”. At the end of the paragraph, this sentence was added: “These animals received average 7.4 g/kg/d dextrose, 3.2 g/kg/d fatty acids and 4 g/kg/d amino acids.”
The paragraph “ PN+GSSG: animals fed PN that is supplemented with 12 μM of GSSG …”, has been modified as follows “PN+GSSG: animals fed PN that is supplemented with 12 μM of GSSG (corresponding to 1.3 mg/kg/d) …”.
C4: The authors provide a significant amount of information on how the genomic DNA was isolated but offer very little in the methods in the section "Methylation analysis". Given this, is a major component of each figure, they should give a more detailed explanation of this protocol.
A: The section (2.2.2) “Methylation analysis” includes 3 subsections (2.2.2.1 to 2.2.2.3) and 2 tables. Thus, the “Methylation analysis” section extends from page 4 to page 6.
C5: The statistics are written in an unusual way. A lot of text is given to "what" the meaning of each difference would represents This really isn't appropriate for the methods section.
A: The following items have been removed:
“ The comparisons were:
- A lack of statistical difference between ON and PN+GSSG, but a difference between [ON + PN+GSSG] and PN, suggests that the effect of PN is caused by in vivo oxidative stress induced by peroxides contaminating PN.
- The absence of a statistical difference between ON and PN will suggest that peroxides from PN do not constitute an additional stress leading to epigenetic change in the gene. However, a difference between [ON + PN] and PN+GSSG will suggest a specific effect of GSSG supplementation.
- A lack of difference between PN and PN+GSSG, but a difference compared to ON, will suggest that PN itself has an effect independent of PN-induced oxidative stress.“
C6: All of the figures could be represented as a single figure with multiple panels.
A: Of course, but a figure with 14 panels would be difficult to integrate into the manuscript. We propose to group the results into 3 figures. The first (figure 2) groups together the results for SOD2 and GPX1. The SOD2 enzyme produces H2O2, while the GPX1 enzyme eliminates it. The second (figure 3) brings together the results for GCLC and GSase. The enzymes GCLC and GSase synthesize glutathione. Figure 4 pools the results for Nrf2 and Keap1. Activation of the transcription factor Nrf2 promotes transcription of the SOD2, GPx1, GCLC and GSase genes. The Keap1 protein inactivates Nrf2.
C7: The presentation of the statistics on the graphs are somewhat confusing. I am not clear what the bar without the asterisks are represent. I would remove those and only show bars connecting significantly different groups. Also, the x-axis should have a line on it.
A: These requested changes have been made.
C8: In the figure legends, typically definitions to abbreviations are at the end of the legend. There is also discussion of results within the legend. I do not think this is necessary.
A: These requested changes about abbreviation have been made. There was no discussion in legends only descriptions of results. These are made so that the figures are autonomous in their understanding.
Comments on the Quality of English Language
C: This paper needs to be revised with a native English speaking editor. There are multiple subject/verb agreement issues and in some cases sentences that do not make any sense.
A: The manuscript has been extensively edited by an English-speaking author.
Thank you for helping us improve the manuscript.
Round 2
Reviewer 2 Report
Comments and Suggestions for Authors
After reading the authors responses, particularly in light of the senior authors retirement and lab closure, I believe the changes that are made are of sufficient quality. I only have three minor changes to the text that are grammatical:
- “Endogenous peroxide (H2O2) is a powerful regulator of several metabolisms based on the oxidation of cysteinyl residues of their proteins.”…Change “metabolism” to metabolic pathways.
- "We hypothesis that early exposure to PN peroxides induces a permanent epigenetic change in the regulation of genes involved in Hâ‚‚Oâ‚‚ metabolism.”…Change to “We hypothesize…”
- “The lack of impact on methylation can be explained by the fact that 95% of the Keap1 promoter region guinea pigs is not established.”…Change to “region in guinea pigs…”
Author Response
Revised Manuscript Nutrients-2819523 entitled: ”Glutathione supplementation prevents neonatal parenteral nutrition induced short- and long-term epigenetic and transcriptional disruptions of hepatic H2O2 metabolism in guinea pigs.”
Authors: Angela Mungala Lengo, Ibrahim Mohamed, Jean-Claude Lavoie *
Answers to the Editor and Reviewer, second round
Reviewer:
Comment: After reading the authors responses, particularly in light of the senior authors retirement and lab closure, I believe the changes that are made are of sufficient quality. I only have three minor changes to the text that are grammatical:
“Endogenous peroxide (H2O2) is a powerful regulator of several metabolisms based on the oxidation of cysteinyl residues of their proteins.”…Change “metabolism” to metabolic pathways.
- "We hypothesis that early exposure to PN peroxides induces a permanent epigenetic change in the regulation of genes involved in Hâ‚‚Oâ‚‚ metabolism.”…Change to “We hypothesize…”
- “The lack of impact on methylation can be explained by the fact that 95% of the Keap1 promoter region guinea pigs is not established.”…Change to “region in guinea pigs…”
Answer: All these corrections have been made. Thank you for your comment.